# Factors Influencing Virtual Art Therapy in Patients with Stroke

**DOI:** 10.3390/brainsci15070736

**Published:** 2025-07-09

**Authors:** Marco Iosa, Roberto De Giorgi, Federico Gentili, Alberto Ciotti, Cristiano Rubeca, Silvia Casolani, Claudia Salera, Gaetano Tieri

**Affiliations:** 1Department of Psychology, Sapienza University of Rome, 00185 Roma, Italy; marco.iosa@uniroma1.it; 2IRCCS Santa Lucia Foundation, 00179 Rome, Italy; gaetano.tieri@unitelmasapienza.it; 3Casa di Cura Nomentana Hospital, 00013 Fonte Nuova, Italy; roberto.degiorgi@nomentanahospital.it (R.D.G.); federico.gentili@nomentanahospital.it (F.G.); francoealberto.roma@gmail.com (A.C.); cristianorubeca@outlook.com (C.R.); casolani.silvia@gmail.com (S.C.); 4Virtual Reality & Digital Neuroscience Lab, Department of Law and Digital Society, Unitelma Sapienza University, 00161 Rome, Italy

**Keywords:** virtual art therapy, Michelangelo effect, neurorehabilitation, psychometry, cultural wellbeing

## Abstract

**Background**: Art therapy was recently administered to stroke patients using immersive virtual reality technology, chosen to provide the illusion of being able to replicate an artistic masterpiece. This approach was effective in improving rehabilitative outcomes due to the so-called Michelangelo effect: patients’ interaction with artistic stimuli reduced perceived fatigue and improved performance. The aim of the present study was to investigate which factors may influence those outcomes (e.g., type of artwork, esthetic valence, perceived fatigue, clinical conditions). **Methods**: An observational study was conducted on 25 patients with stroke who performed the protocol of virtual art therapy (VAT). In each trial, patients were asked to rate the esthetic valence of the artworks and their perceived fatigue, whereas therapists assessed patients’ participation in the therapy (Pittsburgh Rehabilitation Participation Scale, PRPS). Moreover, before and after treatment, patients’ independence in daily living activities (Barthel Index, BI), and their upper limb functioning (Manual Muscle Test, MMT) and spasticity (Ashworth Scale, AS) were measured. **Results**: The after-treatment BI scores depended on the before-treatment BI score (*p* < 0.001) and on the PRPS score (*p* = 0.006), which, in turn, was increased by the subjective esthetic valence (*p* = 0.044). Perceived fatigue is a complex factor that may have influenced the outcomes (*p* = 0.049). **Conclusions**: There was a general effect of art in reducing fatigue and improving participation of patients during therapy. The variability observed among patients mainly depended on their clinical conditions, but also on the esthetic valence given to each artwork, that could also be intertwined with the difficulty of the task. Art therapy has a high potential to improve rehabilitation outcomes, especially if combined with new technologies, but psychometric investigation of the effects of each factor is needed to design the most effective protocols.

## 1. Introduction

Art therapy is recognized by World Health Organization as effective in health promotion, disease prevention, and treatment [1]. In general, protocols are based on art fruition and art production. The former has the advantage of exposing patients to real artistic masterpieces, while the latter engages patients more actively, even though their creations may not be true masterpieces [1]. Both approaches have been proposed for the neurorehabilitation of patients with stroke [2,3,4]. The use of virtual reality (VR) has been suggested as a solution to combining the two approaches, allowing patients to paint on digital canvas in a virtual environment while gaining the illusion of being able to replicate an artistic masterpiece [5]. The VR protocol revealed the so-called Michelangelo effect: with artistic stimuli, both healthy individuals and patients with stroke reported reduced fatigue and showed improved performance compared with when they were asked to color a canvas [5]. Importantly, when combined with conventional therapy in a randomized controlled trial, the virtual art therapy (VAT) protocol proved to be more effective than conventional therapy alone in improving motor functioning of the affected upper limb and enhancing the patient’s independence in daily living activities [6,7]. Also, in randomized controlled trials, the integration of art therapy and digital technologies has already been found effective in improving the mental health of healthy participants [8], individuals with mild cognitive impairment [9], and oncological patients [10]. A review investigated the use of digital telehealth in art therapies for individuals with neurodevelopmental and neurological disorders, reporting psychological, emotional, social, and cognitive benefits [11]. However, most of the reviewed studies were focused on music therapy.

Many art therapy protocols have the advantage of being highly customizable to the needs and preferences of patients. However, this flexibility often results in poorly defined protocols, frequently not based on neuroesthetic principles and lacking a rigorous methodological approach that would support their replicability elsewhere [12]. The aim of this study was to identify the factors affecting the efficacy of this approach in patients with stroke who had been hospitalized in the subacute phase for neurorehabilitation.

The study of prognostic factors in stroke neurorehabilitation is essential for developing personalized treatments while maintaining an approach that is solidly grounded in the context of evidence-based medicine. The main prognostic factors identified as positively influencing the outcome of conventional therapy for stroke include younger age, male gender, lower severity at admission, shorter time between the acute event and the start of therapy, and absence of neuropsychological deficits and comorbidities [13,14,15,16,17,18]. These factors were also confirmed by new approaches involving cluster analyses and artificial intelligence based on neural network analysis [19].

## 2. Materials and Methods

### 2.1. Patient Sample

Twenty-five patients with stroke in the subacute phase, hospitalized for neurorehabilitation, were enrolled in the observational study. The virtual art therapy protocol was administered as routinely performed at the Nomentana Hospital of Rome, and data were collected before, during, and after treatment.

The sample size of 25 patients was determined on the average correlation coefficients computed in previous studies (R = 0.43, 0.58, and 0.6; average value of 0.53) [6,7], selecting an alpha level of 5%, a beta-level of 20%, and thus, a power of 80%. Demographical and clinical characteristics are reported in Table 1.

The study was approved by the Ethical Committee and each patient provided signed informed consent.

### 2.2. Assessment and Therapy

Patients were treated daily, six days per week, for three hours per day. During these sessions, they received physical therapy, cognitive therapy, occupational therapy, and speech therapy, including specific interventions for swallowing, bowel, and bladder dysfunctions based on their clinical needs, and VAT. Virtual art therapy was administered for a period of four weeks, one therapy session per day, three days per week, for a total of 12 sessions.

Patients were clinically assessed before and after the four-week therapy period using the following scales: Barthel Index (BI, a scale ranging from 0 to 100; higher scores indicate greater independence in activities of daily living), Manual Muscle Test (MMT: a scale ranging from 0 to 15, measuring muscle strength in shoulder abduction, elbow flexion, and pinch grip, with higher scores indicating greater strength), and Ashworth Scale (AS: a scale measuring spasticity in the upper limb, ranging from 0, no spasticity, to 5, severe spasticity avoiding passive mobilization). The same clinical assessments were repeated at the end of the four-week therapy period. After each VAT session, the therapist assessed the patient’s participation using the validated Italian version of the Pittsburgh Rehabilitation Participation Scale (PRPS) [20,21]. Additionally, after each trial (painting) within the VAT session, the patient was asked by the therapist to rate, using a numeric rating scale, the objective beauty of the painting (which is assumed to be more related to cultural factors), how much they liked it (which reflects subjective beauty), and the perceived fatigue required to complete the task, as in previous studies [5,6,7].

Each VAT session began with the patient comfortably seated and equipped with a Meta Quest 2 head-mounted display (HMD). The HMD provided a resolution of 1832 × 1920 pixels per eye and operated at a 60 Hz refresh rate. Upon wearing the HMD, patients were immersed in a virtual environment designed to resemble a spacious living room, featuring a virtual white canvas (60 cm × 40 cm; 2400 cm^2^) positioned directly in front of them. Patients used the Meta Quest controller, held in their paretic hand, to interact with the virtual environment and control brush movements. The virtual environment was modeled using 3ds MAX 2018 and implemented within the Unity 2018 game engine software via custom C# scripting. Within the virtual environment, patients saw a virtual spherical brush positioned identically to the handheld controller. They were told that touching the brush to the canvas would create a painting. In fact, the virtual brush removed a series of thin digital mask (white pixels) that covered a preloaded artwork. As the brush moved across the surface, these pixels disappeared, progressively revealing the underlying image of well-known art masterpieces (see Figure 1). The VAT system had already been tested in previous studies, receiving very high usability from both healthy subjects and patients with stroke [5,6].

At the beginning of each single session, the therapist selected the level of task difficulty, customizing it on the basis of the current patient’s motor capabilities: easy level (the canvas is composed of 228 pixels, each with an area of 10.5 cm^2^) or hard level (1178 pixels, each of 2 cm^2^). For each session, patients performed multiple trials, each unveiling a different artwork previously validated in studies focused on esthetic appreciation and beauty perception [5,6,7] (see the Appendix A for the complete list of paintings). Prior to each session, the therapist determined the presentation order of artworks by selecting one out of four possible pre-defined sequences. Sessions were supervised by trained physiotherapists who ensured the accurate execution of movements and assisted patients in maintaining appropriate posture, tailored to their individual rehabilitation objectives. In accordance with standard rehabilitation protocols, therapists were permitted to provide assistance when necessary, and rest periods were offered as required.

### 2.3. Statistical Analysis

Data are reported in terms of mean ± standard deviation (SD), or percentual relative frequency for nominal variables. The Wilcoxon rank test was used to compare parameters before and after treatment. Spearman’s coefficient (R) was computed to assess correlations between parameters. Principal component analysis (PCA) was conducted using Oblimin as the rotational method and using parallel analysis to identify the number of factors. An analysis of variance was conducted on the judgments of the patient reported after each trial about objective beauty, subjective liking, and perceived fatigue, using the presented artwork as the fixed effect and the patient and the difficulty of the task as the random effect, and calculating the partial eta square (η_p_^2^) to quantify the effect size. Finally, the whole group of patients was divided with respect to their PRPS score and whether this was higher or lower than the median value, and the variables were compared between these two subgroups using t-tests. The alpha level was selected as 5% for all the performed analyses. The statistical software used for all the analyses were Jamovi (v. 2.3.21, The Jamovi Project) and IBM SPSS Statistics (version 23, IBM Corp., Armonk, NY, USA).

## 3. Results

Table 1 shows demographical and clinical characteristics of enrolled patients. Figure 2 shows the statistically significant improvements in clinical scores after treatment with respect to baseline (BI: *p* < 0.001; MMT: *p* < 0.001; AS: *p* = 0.023).

The right panel of Figure 2 shows the progressive assessment of the beauty, liking, and fatigue perceived by patients and the PRPS score that was a measure of the patient’s participation in each session as assessed by therapists. On average, VAT was administered over 11 sessions (of the 12 planned) spanning approximately 26 days, with a high level of participation (5.6 out of 6). Patients rated the artistic stimuli as beautiful (7.7 out of 10) and reported low levels of fatigue (2.2 out of 10).

We tested several correlations, in particular to identify the variables significantly related to the clinical outcomes. The clinical scores assessed after treatment showed significant correlations with variables assessed at baseline and with those assessed during each session. In particular, the post-treatment BI score was correlated with pre-treatment measurements of BI (R = 0.647, *p* < 0.001), MMT (R = 0.412, *p* = 0.412), and PRPS (R = 0.538, *p* = 0.006). The post-treatment MMT score was significantly correlated with pre-treatment measurements of MMT (R = 0.488, *p* = 0.013), BI (R = 0.598, *p* = 0.002), and with the number of VAT sessions completed (R = 0.492, *p* = 0.013) and the average perceived fatigue (R = −0.397, *p* = 0.049). Post-treatment Ashworth score was correlated with perceived fatigue (R = −0.399, *p* = 0.048). Also, the more the patient appreciated the artworks, the more he/she actively participated to therapy (R = 0.405, *p* = 0.044).

Because the above correlations could be intertwined, to reduce the dimensionality of the dataset and identify meaningful patterns, we performed a principal component analysis (PCA), the results of which are reported in Table 2. Three factors were identified by this analysis as influencing the data. Factor 1 was mainly loaded by age, spasticity, participation, and esthetic judgments; factor 2 by pre- and post-treatment scores of BI and MMT, post-treatment Ashworth score, and perceived fatigue; factor 3 by numbers of sessions and number of days of therapy.

Figure 3 graphically combines the results of the correlations and the PCA. In this figure, each variable is shown as a circle, and the lines linking two variables represent statistically significant correlations between them (in blue if positive, in red if negative, bold if strong). The variables are horizontally divided into three boxes: those assessed pre-treatment, those assessed during treatment, and those assessed post-treatment. Variables are then vertically positioned accordingly to their main loading on one of the three factors identified by the PCA (which could be interpreted as 1: personal factor, 2: clinical factor, or 3: therapy factor).

Finally, analyses of variance were conducted to analyze the data recorded for each trial in each session. Trial analysis revealed that perceived beauty only depended on the presented artworks (F (29,720) = 9.948, *p* < 0.001, η_p_^2^ = 0.286), and not on task difficulty (F (1,28) = 0.083, *p* = 0.775, η_p_^2^ = 0.003) or on their interaction (F (29,469) = 0.779, *p* = 0.791, η_p_^2^ = 0.046). Mona Lisa of Leonardo was reported as the preferred painting (9.32 out of 10), followed by the Creation of Adam by Michelangelo (9.12), while The Dancers of Matisse was the less preferred one (6.07).

Perceived fatigue was directly influenced by the painting (F (29,835) = 5.179, *p* < 0.001, η_p_^2^ = 0.153), but it was also affected by task difficulty (F (1,27) = 5.290, *p* = 0.030, η_p_^2^ = 0.165), and by their interaction (F (1,27) = 1.916, *p* = 0.003, η_p_^2^ = 0.101). The painting associated with the greatest perceived fatigue was The Annunciation of Leonardo (2.87 out of 10), whereas The Wave of Kanagawa elicited the lowest perceived fatigue (1.49). Surprisingly, the easy task level resulted in slightly, but significantly, more fatiguing (2.18) than the hard level (1.77). However, this difference depended on the painting: The Dancers of Matisse resulted more fatiguing in the easy than in the hard task (3.51 vs. 2.11), whereas The Wave of Kanagawa elicited an opposite pattern (0.86 vs. 1.72, respectively). For some other paintings, the fatigue was similar between the two tasks (such as for the Venus of Botticelli: 1.91 vs. 195, respectively).

To investigate, in greater depth, the effects of participation, which was directly associated with the BI score after therapy, the variables were compared between the group with high participation (13 subjects with a PRPS score higher than the median value of 5.75) and the other patients (12 subjects with a PRPS score ≤ 5.75). The results are reported in Table 3. The patients with higher participation showed a significantly higher BI score after therapy and also expressed greater esthetic appreciation for the paintings. Fatigue was one point lower on the PRPS scale in this group, but this difference was not statistically significant. At admission, these patients had a slightly higher BI score (*p* = 0.048), but with a higher spasticity that approached the statistical significance compared with other subjects (*p* = 0.066).

## 4. Discussion

This study aimed to identify factors influencing the efficacy of virtual art therapy and the relationships among variables assessed before and during therapy. The goal was to understand the role of each factor and their potential interactions within a real clinical setting. Accordingly, an observational study was preferred over an experimental investigation, allowing us to consider some variables as random, instead of fixed factors.

Patients showed a significant improvement in upper limb functioning and independency in activities of daily living, and a reduction in spasticity. The clinical outcomes correlated with the severity of patients at baseline, as well as with some other parameters specific to the treatment: participation to rehabilitation, number of completed VAT sessions, and perceived fatigue.

We can summarize our results as follows: in addition to the conventional prognostic factors for the rehabilitation of patients with stroke, in virtual art therapy the patient’s active participation (which is, in turn, enhanced by the esthetic value attributed to the paintings), their perceived fatigue (a crucial aspect during the treatment), and the number of therapy sessions completed were shown to have a significant role.

Principal component analysis identified three main factors. The first factor was related to age of patients, level of spasticity before treatment, esthetic judgments of stimuli presented during therapy, and active participation during therapy. This factor can be interpreted as related to patient’s engagement during therapy. Greater esthetic appreciation of stimuli during therapy was associated with greater participation in the therapy, and, in turn, to greater independency in activities of daily living. Objective and subjective beauty showed similar scores and both depended on the specific artworks. Some paintings were more appreciated than others, such as those by Leonardo and Michelangelo. Older age seemed to reduce the esthetic appreciation of stimuli. However, since older subjects are usually less affine to VR technology [22], this result may be more related to the use of the VR system, rather than to selection of artistic stimuli. Moreover, the sense of presence, the typical illusion “of being there” elicited by VR, declines with user’s age [23]. Spasticity is known to be a physical limitation affecting the engagement of patient during therapy [24]. The importance of participation in virtual art therapy was further supported by comparing patients who participated more actively in the therapy with those who were less engaged. Despite some minor differences before therapy, the former group showed better outcomes in terms of BI score and a greater appreciation of the paintings. This may suggest an indirect effect of perceived beauty on rehabilitation outcomes: the patient’s active participation could be considered a psychometric construct, with the perceived esthetic value of the stimuli as a formative indicator (a cause) and the clinical outcome as a reflective indicator (an effect). Previous studies have shown that the Michelangelo effect occurs for artistic stimuli, but not for colored paint splotches [5], photographs, or non-artistic paintings [25]. Among artistic stimuli, those judged to be more beautiful enhanced the Michelangelo effect [25]. Based on this evidence, in this study we selected artistic paintings that were generally recognized as beautiful by observers, and indeed, these were rated as highly beautiful by the patients.

The second factor was related to the functional conditions of patients, including BI and MMT scores, both before and after treatment. This finding is in accordance with most studies, which show the clinical scale scores at admission as prognostic factors of the rehabilitation outcome [13,14,15,16,17,18]. The main loading of post-therapy AS scores (but not pre-therapy) was also associated with this factor. Perceived fatigue during therapy was also involved, being higher for patients with lower muscle strength (MMT, both pre- and post-therapy). The significant correlation between fatigue and motor function independence is well documented in the literature [26]. Fatigue may also reduce the patient’s attentional level during therapy [27]. Interestingly, perceived fatigue was not significantly correlated with esthetic appreciation. This result may seem in contrast with the Michelangelo effect, which suggests that the presence of artistic stimuli reduces perceived fatigue. In fact, the Michelangelo effect was defined as the reduction in perceived fatigue and the improvement in a subject’s performance when completing a virtual painting task that recreated artworks, compared to non-artistic images [5]. In a previous study [27], fatigue was perceived as lower for paintings than for photos, but not necessarily for beautiful versus non-beautiful stimuli. In our study, all stimuli were artistic, and according to the Michelangelo effect, a low level of fatigue was perceived (on average: 2.1 out of 10). Therefore, the absence of an effect of perceived beauty on fatigue should not be interpreted as in contrast with previous findings. Indeed, all stimuli represented artistic masterpieces and may have generally reduced fatigue independently of the esthetic valence associated with each artwork.

We found a positive correlation between perceived fatigue during the VAT sessions and spasticity after treatment. This correlation may serve as a warning about the potential risk of increasing spasticity through fatiguing exercises. However, several studies have rejected the hypothesis that fatiguing exercises increase spasticity in stroke [28]. Furthermore, in our study, spasticity was significantly reduced after treatment, suggesting that fatigue may be a consequence observed in patients who retained a certain level of spasticity, and not a cause.

The most surprising result for fatigue was the greater level when using large pixels compared to small pixels. However, it should be noted that the former (i.e., easier task) was used in the first sessions of therapy, when patient’s abilities were still severely affected, whereas only after a certain recovery did the therapists administer tasks with smaller pixels (harder task). This effect was also influenced by the specific artwork. For instance, results showed that The Dancers of Matisse was a less appreciated and a more fatiguing painting when the pixels were large. On the contrary, The Wave of Kanagawa generally required less effort, especially with large pixels. The adaptation of the protocol to the patient’s progressively recovered abilities may have introduced a confounding factor; however, this is a common problem in ecological studies conducted in real clinical settings.

Brain arousal occurring during artistic masterpiece observation is well documented in literature, especially for the activation of the mirror neuron network [29]. Brain areas usually related to motor activities may also be activated by the walkability of a scene [30] or even by the empathetic engagement with the imagined gesture implied by the trace done by the painter into the observer [29,31]. Interestingly, motor area activations occur quite spontaneously, regardless of whether the dynamic scene represents moving people (such as the dancers in Matisse in our study) or moving objects (such as the wave of Kanagawa) [30].

The third factor may be related to therapy compliance: a higher number of sessions was associated with an increased post-therapy MMT score, and the total number of therapy days was dependent on time from stroke onset. Adherence to rehabilitation is an important issue in stroke treatment [32]. The Ashworth score also showed a strong loading on this factor, likely due to its correlation with time. In fact, spasticity can be developed in the first month after the acute event and even increase later, especially in the presence of a severe paresis [33].

This study had several limitations, the first being the small sample size. The second limitation was that not all variables could be randomly distributed among patients, with protocol being adapted to each patient’s clinical condition. For this reason, some variables were considered as random instead of fixed, and this factor was used in the analysis of variance for its robustness to deviation from normal distributions [34]. Then, although the selection of artworks was based on previous studies, not all artistic styles were equally represented. For instance, abstract paintings were not presented to patients, but previous studies show that cortical motor areas are also activated by the observation of static abstract artworks [35]. Therefore, future studies could investigate possible implications of this. Another limitation was that we did not record the patients’ pharmacological treatments; for example, certain medications could have affected the relationship between perceived fatigue and spasticity. However, it is important to specify that none of the patients received botulinum toxin treatment to reduce spasticity, either before or during VAT.

The prognostic factors for stroke rehabilitation are well established in the literature and included the patient’s pre-therapy clinical condition, age, presence of aphasia or unilateral spatial neglect, and a short interval between the acute event and the initiation of rehabilitation [19]. However, no studies have, to date, clarified whether different prognostic factors are involved in traditional art therapy. Our study lacked a control group undergoing traditional art therapy, thus limiting our discussion, as it remains unclear whether our findings pertain specifically to virtual art therapy or can be generalizable to art therapy as a whole. Interestingly there is a growing body of literature in which art therapy has been administered using digital and virtual technologies [8,9,10,11,36,37,38,39,40]

Nevertheless, our results provide valuable insights into additional prognostic factors, such as participation and perceived fatigue, that may be relevant in virtual art therapy, complementing those already recognized in stroke rehabilitation. The role of perceived beauty in therapeutic stimuli should also be explored in future studies by using non-beautiful and non-artistic stimuli in patients with stroke. Currently, there has been only one previous study conducted on healthy participants, which found that paintings were less fatiguing than photographs. Furthermore, beautiful stimuli were not less fatiguing than non-beautiful ones in general, but only for paintings [25].

## 5. Conclusions

The aim of this study was to identify factors influencing the efficacy of virtual art therapy and the relationships among variables in a real clinical context. The first noteworthy result is that outcomes were not directly affected by how much patients appreciated observed artworks. This finding suggests that art is independent of the esthetic valence of specific artworks, indicating that the Michelangelo effect may, likewise, be independent of artwork selection. However, it should be noted that all paintings used in this study were well-known masterpieces generally regarded as beautiful. This may have led to a relatively uniform esthetic valence, potentially limiting its influence on outcomes. Furthermore, an indirect effect of perceived beauty cannot be ruled out; while it may not have exerted a direct influence on the clinical outcome, it could have increased patients’ active participation, which, in turn, may have facilitated improved rehabilitation results.

Nonetheless, patients’ subjective perception of beauty did influence their engagement in therapy, which, in turn, contributed to the recovery of independence in activities of daily living. Interestingly, subjective perception of beauty was not associated with perceived fatigue. Our findings suggest that fatigue is a multifaceted factor associated with both clinical conditions and settings of therapy protocol. On one hand, greater fatigue may reflect higher engagement during task execution; on the other hand, it may indicate physical tiredness and may be associated with upper limb spasticity. Therefore, further studies are necessary to investigate the role of fatigue in the context of art therapy for rehabilitation.

Finally, the most suitable candidates for VAT appear to be stroke patients who are not of advanced age, show adequate upper limb functioning, and present a low risk of developing spasticity.

## Figures and Tables

**Figure 1 brainsci-15-00736-f001:**
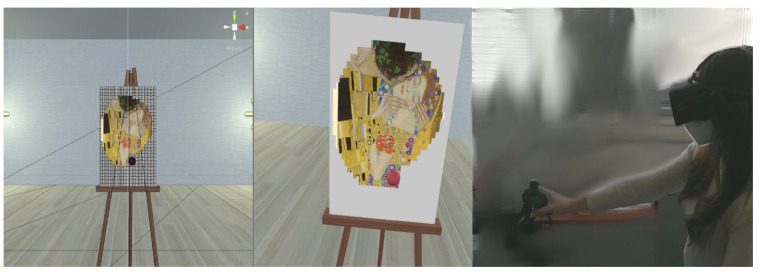
**Left** panel: a schematic representation of the virtual environment in the digital room, with canvas and to-be deleted pixels. **Middle** panel: patients’ view of the subject while painting The Kiss of Klimt. **Right** panel: participant performing the task while wearing the head-mounted display (HMD) and manipulating the virtual canvas with the handheld controller.

**Figure 2 brainsci-15-00736-f002:**
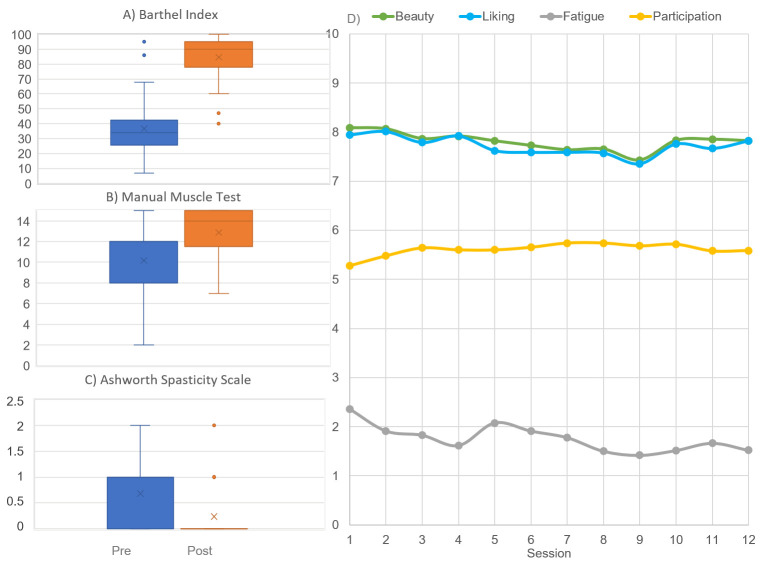
On the left, box and whiskers plot of the clinical scores (in which medians and quartiles are reported for (**A**) Barthel Index, (**B**) Manual Muscle Test, and (**C**) Ashworth Spasticity Scale, together with the means (represented by a cross) pre- (blue) and post- (orange) rehabilitation. On the right, (**D**) the average beauty (green), liking (light blue), and fatigue (grey) perceived by patients, and their participation assessed by therapists (yellow, on a maximum of 6).

**Figure 3 brainsci-15-00736-f003:**
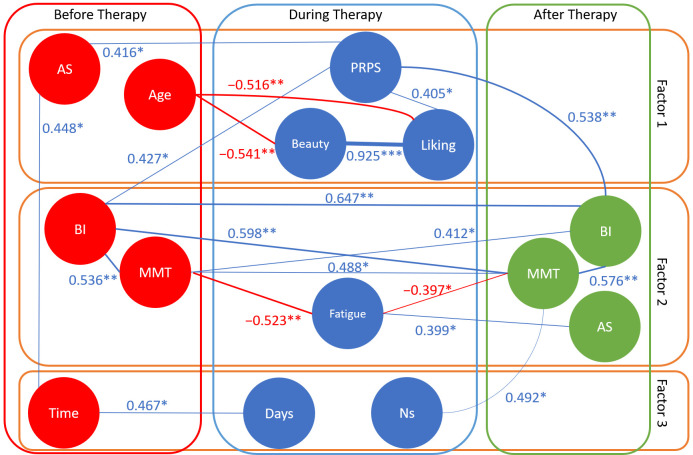
Schematic representation of the statistically significant correlations (in blue if R > 0, in red if R < 0, the bolder the line, the more significant the correlation, R-values are reported close to each relevant line, * *p* < 0.05, ** *p* < 0.01, *** *p* < 0.001) between the variables assessed pre-, during, and post-treatment, grouped according to their main loading on one of the three factors identified by principal component analysis. AS: Ashworth Scale, BI: Barthel Index, MMT: Manual Muscle Test, PRPS: Pittsburgh Rehabilitation Participation Scale, Ns: number of sessions, Time: time between acute event and start of therapy.

**Table 1 brainsci-15-00736-t001:** Data recorded for patients before, during, and after treatment.

Timing	Variables	Mean ± SD or Frequency
Before Therapy	Age (years)	68.1 ± 9.6
	Time from stroke (days)	19.2 ± 8.9
	Gender (F: female; M: male)	F: 40% M: 60%
	Type of Stroke (I: ischemic; H: hemorrhagic)	I: 76% H: 24%
	Affected body side (L: left; R: right)	L: 48% R: 52%
	Barthel Index (BI)	36.7 ± 20.3
	Manual Muscle Test (MMT)	10.2 ± 3.1
	Ashworth Scale (AS)	0.68 ± 0.69
During Therapy	Number of sessions	11.0 ± 2.1
	Length of therapy (days)	25.9 ± 6.5
	PRPS (mean)	5.6 ± 0.4
	VAS perceived fatigue	2.1 ± 1.8
	VAS beauty	7.7 ± 0.8
	VAS liking	7.7 ± 0.8
After Therapy	Barthel Index	84.6 ± 15.9
	MMT	12.9 ± 2.4
	Ashworth Scale	0.2 ± 0.5

**Table 2 brainsci-15-00736-t002:** Results of principal component analysis: loadings of assessed variables on the three factors. In bold the higher load of each variable on a specific factor.

Timing	Variable	Factor 1	Factor 2	Factor 3
Before Therapy	Age	**−0.774**	−0.053	−0.188
	Time	0.220	0.127	0.715
	BI	0.316	**0.494**	0.020
	MMT	0.156	**0.808**	−0.180
	AS	**0.487**	−0.071	0.474
During Therapy	N° sessions	−0.161	0.125	**0.752**
	N° days	−0.107	−0.089	**0.890**
	PPRS	**0.699**	0.082	0.105
	Beauty	**0.857**	−0.052	−0.173
	Liking	**0.848**	0.052	−0.163
	Fatigue	−0.099	**−0.483**	0.302
After Therapy	BI	0.068	**0.866**	0.082
	MMT	−0.090	**0.889**	0.129
	AS	0.417	**−0.628**	0.034

**Table 3 brainsci-15-00736-t003:** Mean ± standard deviations of the variables when subjects were divided into two groups with respect to the median PRPS score, with the relevant *p*-values.

Assessment Timing	Variable	High Participation PRPS > 5.75 (N = 13)	Medium Participation PRPS ≤ 5.75 (N = 12)	*p*
Before Therapy	Age (years)	66.1 ± 10.3	70.3 ± 8.8	0.288
	Days from stroke	19.3 ± 10.8	19.0 ± 6.1	0.937
	Ashworth score	0.9 ± 0.8	0.4 ± 0.5	0.066
	MMT score	11.2 ± 2.1	9.1 ± 3.8	0.099
	BI score	44.3 ± 25.2	28.4 ± 8.1	0.048
During Therapy	Participation PRPS	5.9 ± 0.1	5.2 ± 0.4	<0.001
	N° sessions	11.1 ± 2.0	10.9 ± 2.2	0.851
	N° days	25.3 ± 5.6	26.6 ± 7.5	0.634
	Fatigue	1.6 ± 1.3	2.6 ± 2.2	0.182
	Beauty	8.0 ± 0.8	7.4 ± 0.7	0.070
	Liking	8.0 ± 0.7	7.3 ± 0.6	0.011
After Therapy	Ashworth score	0.2 ± 0.4	0.2 ± 0.6	0.929
	MMT score	13.5 ± 2.0	12.2 ± 2.7	0.161
	BI score	93.1 ± 7.0	75.3 ± 17.9	0.003

## Data Availability

Data supporting the findings of this study are available at the Open Science Framework at the following link: https://osf.io/p9jtc/.

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
