# Peer review of "Factors Influencing Virtual Art Therapy in Patients with Stroke"

_brainsci, 2025, doi:10.3390/brainsci15070736_

Round 1

Reviewer 1 Report

Comments and Suggestions for Authors

I think this is a very interesting study, but to attribute this work to the Michelan effect I think more evidence is needed。As  the author's conclusion clearly states that the completion of the task has nothing to do with whether the painting is beautiful or not, and if so, is the patient's motivation for completing the task the Michelangelo effect or is it just to fully embody the image at the bottom? That is, whether there is an emotional factor in the process, because art is related to feelings. I suggest replacing the painting at the bottom with a geometric image to see if the patient can achieve the same effect, if the same effect may be a task-oriented effect, if it is not as good as the painting, it may be related to the Michelangelo effect.

Author Response

AUTHORS: We would like to thank the Reviewer for the general positive comment about our study. We have carefully taken into account his/her suggestions for improving our manuscript in its revised version as shown in our following point-by-point responses, in which we have also reported (between apices and in italic style) the new/modified parts of the resubmitted manuscript.

REVIEWER: As the author's conclusion clearly states that the completion of the task has nothing to do with whether the painting is beautiful or not, and if so, is the patient's motivation for completing the task the Michelangelo effect or is it just to fully embody the image at the bottom?

That is, whether there is an emotional factor in the process, because art is related to feelings. I suggest replacing the painting at the bottom with a geometric image to see if the patient can achieve the same effect, if the same effect may be a task-oriented effect, if it is not as good as the painting, it may be related to the Michelangelo effect.

AUTHORS: Thanks for this comment. Previous literature partially investigated if the Michelangelo effect was more related to art or the beauty of the stimuli. It is difficult for us to add new tests on the patients with other stimuli (the geometrical images), but we could refer to previous studies in which the canvas were painted with colour splotches, or non-artistic paintings were used. Then, we cannot rule out an effect of beauty, there is the possibility of an indirect effect. To deeply investigate this aspect (being not possible to follow at the moment the nice suggestion of the reviewer to use control geometrical stimuli), we have added to the revised version of our manuscript a comparison of the patients who showed a high active participation with those who had not. We have added the following paragraphs in Results and Discussion, with a new Table 3.

Results:

“To investigate in greater depth the effects of participation, which was directly associated with the BI-score after therapy, the variables were compared between the group with high participation (13 subjects with a PRPS-score higher than the median value of 5.75) and the other patients (12 subjects with a PRPS-score ≤ 5.75). The results are reported in Table 3. The patients with higher participation showed a significantly higher BI-score after therapy and also expressed greater aesthetic appreciation for the paintings. Fatigue was one point lower on the PRPS scale in this group, but this difference was not statistically significant. At admission, these patients had a slightly higher BI-score (p=0.048), but with a higher spasticity that approached the statistical significance compared to other subjects (p=0.066).”

Table 3. Mean ± standard deviations of the variables when subjects where divided into two groups with respect to the median PRPS-score, with the relevant p-values.

Assessment Timing

Variable

High Participation

PRPS>5.75

(N=13)

Medium Participation

PRPS≤5.75

(N=12)

p

Before

Therapy

Age (years)

66.1±10.3

70.3±8.8

0.288

Days from Stroke

19.3±10.8

19.0±6.1

0.937

Ashworth score

0.9±0.8

0.4±0.5

0.066

MMT score

11.2±2.1

9.1±3.8

0.099

BI score

44.3±25.2

28.4±8.1

0.048

During

Therapy

Participation PRPS

5.9±0.1

5.2±0.4

<0.001

N° sessions

11.1±2.0

10.9±2.2

0.851

N° days

25.3±5.6

26.6±7.5

0.634

Fatigue

1.6±1.3

2.6±2.2

0.182

Beauty

8.0±0.8

7.4±0.7

0.070

Liking

8.0±0.7

7.3±0.6

0.011

After

Therapy

Ashworth score

0.2±0.4

0.2±0.6

0.929

MMT score

13.5±2.0

12.2±2.7

0.161

BI score

93.1±7.0

75.3±17.9

0.003

Discussion:

“The importance of participation in virtual art therapy was further supported by the comparison between patients who participated more actively in the therapy and those who were less engaged. Despite some minor differences before therapy, the former group showed better outcome in terms of BI-score and greater appreciation of the paintings. This may suggest an indirect effect of perceived beauty on rehabilitation outcomes: the patient’s active participation could be considered a psychometric construct, with the perceived aesthetic value of the stimuli as a formative indicator (a cause) and the clinical outcome as a reflective indicator (an effect). Previous studies have shown that the Michelangelo effect occurs for artistic stimuli, but not for colored paint splotches [5], photographs or non-artistic paintings [25]. Among artistic stimuli, those judged to be more beautiful enhanced the Michelangelo effect [25]. Based on these evidences, in this study we selected artistic paintings that are generally recognized as beautiful by observers, and indeed they were rated as highly beautiful by the patients.”

We have also added this sentence in the Conclusions:

Furthermore, an indirect effect of perceived beauty cannot be ruled out: while it may not have exerted a direct influence on the clinical outcome, it could have increased patients’ active participation, which in turn may have facilitated improved rehabilitation results.”

Reviewer 2 Report

Comments and Suggestions for Authors

The aim of this study was to identify the factors influencing the efficacy of virtual art therapy and the relationships among variables involved in the Michelangelo effect in a real clinical context. Overall, the study is well designed and executed; although the sample size is not very large, it may be sufficient depending on the type of pathology addressed. The timing of the rehabilitation, the number of sessions, and the assessments conducted are all appropriate.

However, I would like to highlight some areas for improvement, particularly regarding the statistical section. The information presented in Table 2 and Figure 2, which describe the evolution of users undergoing treatment, is not very clear. Even after several readings, I still do not fully understand these sections. Therefore, I suggest revising them to make them more specific and easier to interpret, clearly detailing the statistical tests used in each case and explaining how significant relationships between variables were obtained. Additionally, it would be advisable to accompany the results with a more direct and accessible interpretation for the reader, thus facilitating a better understanding of the main findings of the study.

Author Response

AUTHORS: We would like to thank the Reviewer for the general positive comment about our study. We have carefully taken into account his/her suggestions for improving our manuscript in its revised version as shown in our following point-by-point responses in which we have also reported (between apices and in italic style) the new/modified parts of the resubmitted manuscript.

REVIEWER: However, I would like to highlight some areas for improvement, particularly regarding the statistical section. The information presented in Table 2 and Figure 2, which describe the evolution of users undergoing treatment, is not very clear. Even after several readings, I still do not fully understand these sections. Therefore, I suggest revising them to make them more specific and easier to interpret, clearly detailing the statistical tests used in each case and explaining how significant relationships between variables were obtained. Additionally, it would be advisable to accompany the results with a more direct and accessible interpretation for the reader, thus facilitating a better understanding of the main findings of the study.

AUTHORS: Thanks for this comment, we have carefully revised our manuscript to improve its clarity. First of all, we have added in the results section a new figure to immediately show how the main variables of this study changed between pre- and post-treatment (on the left) and along the twelve planned sessions (on the right). Then we have rewritten the results leading to the ex Figure 2 (now 3) reporting the results of correlations and Principal Component Analysis.

Hence, in the main text we have described the new figure 2:

“Figure 2 shows the improvements obtained in terms of clinical scores that resulted statistically significant after treatment with respect to baseline (BI: p<0.001; MMT: p<0.001; AS: p=0.023).

The right panel of Figure 2 shows the progressive assessment of the beauty, liking and fatigue perceived by patients and the PRPS-score that was a measure of the patient’s participation to each session assessed by therapists.”

The new Figure 2 is the following (with its caption):

“Figure 2. On the left box whiskers plot of the clinical scores (in which medians and quartiles are reported for A: Barthel Index, B: Manual Muscle Test, C: Ashworth Spasticity Scale, together with the means represented by a cross) pre- (blue) and post- (orange) rehabilitation. On the right (D panel) the average beauty (green), liking (light blue), and fatigue (grey) perceived by patients, and their participation assessed by therapists (yellow, on a maximum of 6).”

Then, we have modified the following paragraphs of the Results for better explaining the performed correlations and principal component analysis:

We tested several correlations, in particular to identify the variables significantly related to the clinical outcomes. (…)

Because the above correlations could be intertwined, to reduce the dimensionality of the dataset and identify meaningful patterns, we performed a Principal Component Analysis (PCA), the results of which are reported in Table 2. Three factors were identified by this analysis as influencing the data. Factor 1 was mainly loaded by age, spasticity, participation and aesthetic judgments; factor 2 by pre- and post-treatment scores of BI, MMT, post-treatment Ashworth score, and perceived fatigue; factor 3 by numbers of sessions and number of days of therapy. 

Figure 3 graphically combined the results of the correlations and the PCA. In this figure, each variable is shown as a circle, and the lines linking two variables represent statistically significant correlations between them (in blue if positive, in red if negative, bold if strong). The variables are horizontally divided into three boxes: those assessed pre-treatment, those assessed during treatment, and those assessed post-treatment. Variables are then vertically positioned accordingly to their main loading on one of the three factors identified by the PCA (which could be interpreted as 1: personal factor, 2: clinical factor, 3: therapy factor).”

We hope that now the meaning of these results are more clearly explained. Then, we have also added the following paragraph at the beginning of Discussion to summarize and clarify the most important findings:

“We could summarize our results as follows: in addition to the conventional prognostic factors for the rehabilitation of patients with stroke, in virtual art therapy the patient’s active participation (which is, in turn, enhanced by the aesthetic value attributed to the paintings), their perceived fatigue (a crucial aspect during the treatment) and the number of therapy sessions completed showed to have a significant role.”

Reviewer 3 Report

Comments and Suggestions for Authors

This paper investigates the factors influencing the efficacy of VAT in stroke patients. While the study contributes to the field of neurorehabilitation, several issues need to be sorted out.

  1. The authors mention the “Michelangelo effect” to explain VAT's gains, but the evidence presented in the study is not completely convincing. The study does not provide an in depth exploration of the underlying neural mechanisms that could provide an explanation for this phenomenon.
  2. The lack of a control group (e.g., traditional art therapy) undermines causal inference.
  3. The assessment tools are predominantly based on subjective self-report, and therefore can be vulnerable to bias. For instance, the high aesthetic score of Mona Lisa might depend more on cultural bias than on neural activity.
  4. The authors fail to consider whether fatigue-spasticity correlations might be related to effects of medication response rather than specific treatment-related alterations.
  5. Tasks were therapist-determined difficulty (“easy” vs. “hard”) based on patient capacity and were provided sequentially. This introduces very severe confounding: The difficulty level is confounded with phase of the treatment and baseline severity of impairment.
  6. A 60Hz refresh rate is low for HMDs and may contribute to discomfort/fatigue, especially in a neuro population.

Author Response

AUTHORS: We would like to thank the Reviewer for the general positive comment about our study. The Reviewer highlighted some weak points of our study, and before analysing them point-by-point we would say that we agree with the reviewer, and reported them as limits of our study. Some of them were strictly related to the fact that it is a clinical study, not performed in a laboratory in which many conditions could be controlled. Anyway, we have carefully taken into account his/her suggestions for improving our manuscript in its revised version as shown in our following point-by-point responses in which we have also reported (between apices and in italic style) the new/modified parts of the resubmitted manuscript.

REVIEWER: The authors mention the “Michelangelo effect” to explain VAT's gains, but the evidence presented in the study is not completely convincing. The study does not provide an in depth exploration of the underlying neural mechanisms that could provide an explanation for this phenomenon.

AUTHORS: This is a good point, but at the best of our knowledge there are no studies investigating the neural networks underlying the Michelangelo effect. There are only some hypotheses connecting it to the fMRI studies in which subjects observed artistic stimuli and the result is a brain arousal, even involving motor cortex. However, we have got the point of the Reviewer, and we understood that the first sentence of the Discussion (that was “The study aimed to identify factors influencing the efficacy of virtual art therapy and the relationships among variables involved in the Michelangelo effect in stroke patients undergoing neurorehabilitation during the subacute phase”) can be seen as an overstatement. For these reason, we have changed it as follows:

This study aimed to identify factors influencing the efficacy of virtual art therapy and the relationships among variables assessed before and during therapy”.

This change is very important, because it moves the focus of the discussion on the Virtual Art Therapy, without forcing a link with the Michelangelo Effect. We also modified the first sentence of Conclusions (that was “The aim of this study was to identify factors influencing the efficacy of virtual art therapy and the relationships among variables involved in the Michelangelo effect in a real clinical context.”) as follows:

“The aim of this study was to identify factors influencing the efficacy of virtual art therapy and the relationships among variables in a real clinical context”

Then, in order to avoid overstatement, we have also deleted this sentence because not fully demonstrated by data “Finally, while the specific artworks used play a partial role, they do not fully account for the effectiveness of the Michelangelo effect in enhancing patient performance.”

REVIEWER: The lack of a control group (e.g., traditional art therapy) undermines causal inference.

AUTHORS: The Reviewer is right, we did not plan to integrate the study with a control group performing traditional art therapy. It is difficult also to discuss our prognostic factors in comparisons to those of traditional art therapy because, despite the prognostic factors in stroke rehabilitation are well established, at the best of our knowledge there are no studies investigating prognostic factors in traditional art therapy. We have added the following paragraph in our Discussion:

“The prognostic factors for stroke rehabilitation are well established in the literature and included the patient’s pre-therapy clinical condition, age, presence of aphasia or unilateral spatial neglect, and a short interval between the acute event and the initiation of rehabilitation [19]. However, no studies to date clearly clarified whether different prognostic factors are involved in traditional art therapy. Our study lacks of a control group undergoing traditional art therapy, thus limiting our discussion, as it remains unclear whether our findings pertain specifically to virtual art therapy or can be generalizable to art therapy as a whole. Nevertheless, our results provide valuable in-sights into additional prognostic factors, such as participation and perceived fatigue, that may be relevant in virtual art therapy, complementing those already recognized in stroke rehabilitation. The role of perceived beauty in therapeutic stimuli should be explored in future studies also using non-beautiful and non-artistic stimuli in patients with stroke. Currently, there is only one previous study conducted on healthy participants, which found that paintings were less fatiguing than photographs. Furthermore, beautiful stimuli did not result less fatiguing than non-beautiful ones in general, but only for paintings [25].”

However, to increase the solidity of our hypothesized causal inferences by having a comparisons between patients, in the revised version of our manuscript we have added a new analysis in which the whole sample was divided in patients with higher (than median value) and lower participation (PRPS-score), and the assessed variables were compared between the two groups. We have added the following main text in the results and in discussion with a new Table 3, as follows:

Results:

“To investigate in greater depth the effects of participation, which was directly associated with the BI-score after therapy, the variables were compared between the group with high participation (13 subjects with a PRPS-score higher than the median value of 5.75) and the other patients (12 subjects with a PRPS-score ≤ 5.75). The results are reported in Table 3. The patients with higher participation showed a significantly higher BI-score after therapy and also expressed greater aesthetic appreciation for the paintings. Fatigue was one point lower on the PRPS scale in this group, but this difference was not statistically significant. At admission, these patients had a slightly higher BI-score (p=0.048), but with a higher spasticity that approached the statistical significance compared to other subjects (p=0.066).”

Table 3. Mean ± standard deviations of the variables when subjects where divided into two groups with respect to the median PRPS-score, with the relevant p-values.

Assessment Timing

Variable

High Participation

PRPS>5.75

(N=13)

Medium Participation

PRPS≤5.75

(N=12)

p

Before

Therapy

Age (years)

66.1±10.3

70.3±8.8

0.288

Days from Stroke

19.3±10.8

19.0±6.1

0.937

Ashworth score

0.9±0.8

0.4±0.5

0.066

MMT score

11.2±2.1

9.1±3.8

0.099

BI score

44.3±25.2

28.4±8.1

0.048

During

Therapy

Participation PRPS

5.9±0.1

5.2±0.4

<0.001

N° sessions

11.1±2.0

10.9±2.2

0.851

N° days

25.3±5.6

26.6±7.5

0.634

Fatigue

1.6±1.3

2.6±2.2

0.182

Beauty

8.0±0.8

7.4±0.7

0.070

Liking

8.0±0.7

7.3±0.6

0.011

After

Therapy

Ashworth score

0.2±0.4

0.2±0.6

0.929

MMT score

13.5±2.0

12.2±2.7

0.161

BI score

93.1±7.0

75.3±17.9

0.003

Discussion:

“The importance of participation in virtual art therapy was further supported by the comparison between patients who participated more actively in the therapy and those who were less engaged. Despite some minor differences before therapy, the former group showed better outcome in terms of BI-score and greater appreciation of the paintings. This may suggest an indirect effect of perceived beauty on rehabilitation outcomes: the patient’s active participation could be considered a psychometric construct, with the perceived aesthetic value of the stimuli as a formative indicator (a cause) and the clinical outcome as a reflective indicator (an effect). Previous studies have shown that the Michelangelo effect occurs for artistic stimuli, but not for colored paint splotches [5], photographs or non-artistic paintings [25]. Among artistic stimuli, those judged to be more beautiful enhanced the Michelangelo effect [25]. Based on these evidences, in this study we selected artistic paintings that are generally recognized as beautiful by observers, and indeed they were rated as highly beautiful by the patients.”

REVIEWER: The assessment tools are predominantly based on subjective self-report, and therefore can be vulnerable to bias. For instance, the high aesthetic score of Mona Lisa might depend more on cultural bias than on neural activity.

AUTHORS: Well, it is true that most of the used variables are self-reported or assessed by clinician. However, it should consider that the Pittsburgh Rehabilitation Participation Scale is a validated tool for assessing participation, Numeric Rating Scale is usually used for the beauty perception. About the aesthetic score, we perfectly agree with the possibility that some judgments are more related to cultural factors and some other to personal factors, for this reason we have differently asked to assess objective beauty and subjective beauty. We specified it, adding the following sentence in the methods section:
“After each VAT session, the therapist assessed the patient’s participation using the validated Italian version of the Pittsburgh Rehabilitation Participation Scale (PRPS) [20,21]. Additionally, after each trial (painting) within the VAT session, the patient was asked by the therapist to rate, using a Numeric Rating Scale, the objective beauty of the painting (which is assumed to be more related to cultural factors), how much they liked it (which reflects subjective beauty), and the perceived fatigue required to complete the task, as done in previous studies [5-7].”

REVIEWER: The authors fail to consider whether fatigue-spasticity correlations might be related to effects of medication response rather than specific treatment-related alterations.

AUTHORS: To take into account this comment, we have added the following sentence in the section about the limits of our study

“Another limitation is that we did not record the patients’ pharmacological treatments; for example, certain medications could have affected the relationship between perceived fatigue and spasticity. However, it is important to specify that none of the patients received botulinum toxin treatment to reduce spasticity, either before or during VAT.”

REVIEWER: Tasks were therapist-determined difficulty (“easy” vs. “hard”) based on patient capacity and were provided sequentially. This introduces very severe confounding: The difficulty level is confounded with phase of the treatment and baseline severity of impairment.

AUTHORS: That is true, but at the same time, it makes the study more ecological. The personalization/adaptation of the therapy is key-point of rehabilitation, but it introduces a confounding when the efficacy of a protocol should be assessed. We agree that we have poorly commented this aspect in Discussion, so in the revised manuscript we have added the following sentence:
“The adaptation of the protocol to the patient’s progressively recovered abilities may have introduced a confounding factor; however, this is a common problem in ecological studies conducted in real clinical settings.”

REVIEWER: A 60Hz refresh rate is low for HMDs and may contribute to discomfort/fatigue, especially in a neuro population.

AUTHORS: Patients did not report discomfort, furthermore, in previous studies using the same device, the usability was assessed as very high. We specified that adding the following sentence in Methods section:

“The VAT system had already been tested in previous studies, receiving very high usability from both healthy subjects and patients with stroke [5,6].”

Reviewer 4 Report

Comments and Suggestions for Authors

The manuscript investigates the psychological and clinical determinants of efficacy in immersive VR–based art therapy for stroke patients. The paper lacks a comprehensive analysis of the state of the art to clearly show the added contribution of their work. The current introduction and discussion sections do not fully integrate recent multicenter randomized trials, systematic reviews, and other digital‐rehabilitation studies, thereby limiting the precise novelty of this observational work.

The reference list is overly concentrated on the authors’ own contributions—11 of the 29 citations originate from Iosa, Paolucci, Antonucci, and their collaborators—raising concerns of self‐citation bias and underrepresentation of independent research.

Figure 1, though aesthetically pleasing, relies exclusively on line‐weight to convey correlation magnitude and statistical significance. To facilitate immediate interpretation, each connection should be annotated with its exact correlation coefficient and, where space permits, the corresponding p-value.

In the Methods section, authors should specify the statistical software and packages employed.

Author Response

AUTHORS: We failed to find a large body of literature about the use of virtual art therapy, in terms of art therapy administered by virtual reality systems. Our researches on Pubmed identified few papers with the string of keywords that we used. For example:

"Virtual Reality" AND "art therapy" AND rehabilitation AND stroke: 4 articles, 3 of them already cited in our manuscript and 1 in Russian language

"Virtual Reality" AND "art therapy" AND RCT: 0 articles

"Virtual Reality" AND "art therapy" AND Review: 9 articles, most of them not related to rehabilitation

Anyway, we have added in the revised version of our manuscript the following paragraphs to integrate the state of the art.

“Importantly, when combined with conventional therapy in a randomized controlled trial, the Virtual Art Therapy (VAT) protocol proved to be more effective than conventional therapy alone in improving motor functioning of the affected upper limb and enhancing patient’s independency in activities of daily living [6,7]. The integration of art therapy and digital technologies has already been found effective in randomized controlled trials also for improving the mental health of healthy participants [8], individuals with mild cognitive impairment [9], and oncological patients [10]. A review investigated the use of digital telehealth in art therapies for individuals with neurodevelopmental and neurological disorders, reporting psychological, emotional, social and cognitive benefits [11]. However, most of the reviewed studies were focused on music therapy. 

Many art therapy protocols have the advantage of being highly customizable on the needs and preferences of patients. However, this flexibility often results in poorly defined protocols, frequently not based on neuroaesthetic principles and lacking a rigorous methodological approach that would support their replicability elsewhere [12]. The aim of this study was to identify the factors affecting the efficacy of this approach in patients with stroke hospitalized in the subacute phase for neurorehabilitation.”

Adding the following articles:

Peng, M. L., Monin, J., Ovchinnikova, P., Levi, A., & McCall, T. (2024). Psychedelic Art and Implications for Mental Health: Randomized Pilot Study. JMIR formative research, 8, e66430. https://doi.org/10.2196/66430

Cao, Y., Yin, H., Hua, X., Bi, S., & Zhou, D. (2025). Effects of artificial intelligence and virtual reality interventions in art therapy among older people with mild cognitive impairment. Australasian journal on ageing, 44(1), e70006. https://doi.org/10.1111/ajag.70006

McCabe C, Roche D, Hegarty F, McCann S. “Open Window”: a randomized trial of the effect of new media art using a virtual window on quality of life in patients’ experiencing stem cell transplantation. Psychooncology. 2013;22(2):330–7. doi: 10.1002/pon.2093.

Reitere, Ē., Duhovska, J., Karkou, V., & Mārtinsone, K. (2024). Telehealth in arts therapies for neurodevelopmental and neurological disorders: a scoping review. Frontiers in psychology, 15, 1484726. https://doi.org/10.3389/fpsyg.2024.1484726

Other articles have been added in Discussion to briefly introduce the increasing body of literature focused on digital and virtual art therapies:

Kim, J., and Chung, Y. J. (2024). A single case study of digital art therapy for a child with ADHD using the metaverse platform. Arts Psychother. 89:102146. doi: 10.1016/j.aip.2024.102146

Weinberg, D. J. (1985). The potential of rehabilitative computer art therapy for the quadriplegic, cerebral vascular accident and brain trauma patient. Art Ther. 3, 66–72. doi: 10.1080/07421656.1985.10758788

Lian, N. W. (2023). Application of virtual reality technology and its impact on digital health in healthcare industry. J. Commer. Biotechnol. 27:4. doi: 10.5912/jcb1320

Zubala, A., Kennell, N., and Hackett, S. (2021). Art therapy in the digital world: an integrative review of current practice and future directions. Front. Psychol. 12:595536. doi: 10.3389/fpsyg.2021.600070

Kuleba, B. (2008). The Integration of Computerized Art-Making as a Medium in Art Therapy Theory and Practice. Master’s thesis, Drexel University, Philadelphia, PA.

If the Reviewer would suggest other articles or reviews about the art therapy administered using virtual reality in stroke rehabilitation that we could integrate in our manuscript, we will be happy to cite and comment them.

REVIEWER: The reference list is overly concentrated on the authors’ own contributions—11 of the 29 citations originate from Iosa, Paolucci, Antonucci, and their collaborators—raising concerns of self‐citation bias and underrepresentation of independent research.

AUTHORS: This is strange for us that the Reviewer consider the citations of Paolucci or Antonucci as self-citations, being these researchers not listed as authors of our manuscript. If we referred only by articles in which one of the authors of the present study is listed as author, according to the definition of self-citation, there are 8 references on 40, that we think is quite normal being these articles those in which we developed, tested and used the virtual reality system.

REVIEWER: Figure 1, though aesthetically pleasing, relies exclusively on line‐weight to convey correlation magnitude and statistical significance. To facilitate immediate interpretation, each connection should be annotated with its exact correlation coefficient and, where space permits, the corresponding p-value.

AUTHORS: The Reviewer is right, it was a problem of space. Anyway, we have now modified the Figure 1 including R-values and asterisks representing the p-values (*p<0.05, **p<0.01, ***p<0.001) as required.

REVIEWER: In the Methods section, authors should specify the statistical software and packages employed.

AUTHORS: We have added this information in the revised version of our manuscript writing:
“The statistical software used for all the analyses were Jamovi (v. 2.3.21, The Jamovi Project) and IBM SPSS Statistics (version 23, IBM Corp., Armonk, NY, USA).”

Round 2

Reviewer 1 Report

Comments and Suggestions for Authors

I think the author's explanation and revision of the article this time are realistic, and the relevant content and results of the supplement have a more objective explanation of the previous results, and it is recommended to publish it.

Comments on the Quality of English Language

no 

Reviewer 3 Report

Comments and Suggestions for Authors

The author has responded to the questions raised and it is recommended for publication.

Reviewer 4 Report

Comments and Suggestions for Authors

The authors have addressed the reviewers’ concerns by incorporating additional references and enriching their manuscript with more detailed descriptions of both the experimental procedures and the resulting findings.